

# Hydrodynamic modes of homogeneous and isotropic fluids

Jan de Boer[1], Jelle Hartong[1], Niels A. Obers[2], Watse Sybesma[3] and Stefan Vandoren[3]

**1** Institute for Theoretical Physics and Delta Institute for Theoretical Physics,
University of Amsterdam, Science Park 904, 1098 XH Amsterdam, The Netherlands
**2** The Niels Bohr Institute, Copenhagen University,
Blegdamsvej 17, DK-2100 Copenhagen Ø, Denmark
**3** Institute for Theoretical Physics and Center for Extreme Matter and Emergent Phenomena,
Utrecht University, 3508 TD Utrecht, The Netherlands

## Abstract

Relativistic fluids are Lorentz invariant, and a non-relativistic limit of such fluids leads to the well-known Navier–Stokes equation. However, for fluids moving with respect to a reference system, or in critical systems with generic dynamical exponent $z$, the assumption of Lorentz invariance (or its non-relativistic version) does not hold. We are thus led to consider the most general fluid assuming only homogeneity and isotropy and study its hydrodynamics and transport behaviour. Remarkably, such systems have not been treated in full generality in the literature so far. Here we study these equations at the linearized level. We find new expressions for the speed of sound, corrections to the Navier–Stokes equation and determine all dissipative and non-dissipative first order transport coefficients. Dispersion relations for the sound, shear and diffusion modes are determined to second order in momenta. In the presence of a scaling symmetry with dynamical exponent $z$, we show that the sound attenuation constant depends on both shear viscosity and thermal conductivity.

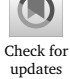

# 1   Introduction

Fluids and gasses are all around us. Their descriptions arise as a universal limit of finite temperature systems when we consider these on sufficiently long length and time scales so that they relax to an approximate thermal equilibrium. The universality of fluid dynamics is what makes it a very powerful and useful framework, providing an effective description based on symmetry principles for wide classes of systems.

Often, one is dealing with a system in which the fluid particles move in a certain medium, for instance electrons in an atomic lattice, or particles swimming collectively in another fluid, birds flocking through the air, etc. The existence of such a medium defines a preferred frame, and this becomes important when the interaction between the fluid particles and the medium cannot be ignored. To get an effective hydrodynamic description for the fluid particles, one should integrate out the degrees of freedom of the medium. Doing so, the dynamics of the fluid particles will in general not have any relativistic symmetry, the Lorentz symmetry, or its non-relativistic version, the Galilei symmetry. In particular, this means that the effective fluid description will not have a boost symmetry relating all inertial frames.

These considerations motivated us to formulate a new effective theory of hydrodynamics in the absence of boost symmetry. The first steps at the level of perfect fluids were taken in [1]. In this paper we take the next important step and develop a linearized theory of hydrodynamic modes for homogeneous and isotropic systems, i.e. systems with only time and space translation symmetries, and spatial rotation symmetry respectively[1]. The extension to the full non-linear level will appear in Ref. [4].

The nature of a boost symmetry or lack thereof has profound implications for the dynamics of fluids[2]. Indeed, as shown in [1], already at the perfect fluid level there are novel expressions for the speed of sound, which were furthermore illustrated in that paper by considering the thermodynamics of a gas of free Lifshitz particles (both classical and quantum). As we will show in the present work by considering the hydrodynamic modes, there will also be new first-order transport coefficients, both dissipative and non-dissipative. They contribute to the transport properties such as sound attenuation and therefore leave an observable imprint on transport phenomena, which can potentially be measured. Also, the effect of broken boost symmetry produces corrections to the Navier-Stokes equation, which we derive at linearized order.

Fluid dynamics studies global conservation laws in a large wavelength approximation. These conservation laws are energy and momentum conservation and possibly a set of $U(1)$

---

[1]The study of hydrodynamics for anisotropic systems has been studied in several places, e.g. in [2,3].

[2]Hydrodynamics of systems without boosts has been addressed previously (see e.g. [5,6] and [7–9]) but our starting point, the perfect fluid thermodynamics [1], differs from these works.

charge conservation laws such as particle number, electric charge, baryon number etc.. Such conservation laws are a consequence of time and space translational symmetries (and possibly some global $U(1)$ symmetries) of the underlying theory of which we are making a fluid approximation. To apply the rules of fluid dynamics we thus do not need to impose any additional symmetries such as boost invariance.

In [1] it was shown that whenever a scale invariant system admits a fluid description with a dynamical exponent $z \neq 1, 2$ it cannot be boost invariant. When $z = 1$ the fluid can have Lorentz boost symmetries and when $z = 2$ the fluid can have Galilean boost symmetries[34]. Since values of $z \neq 1, 2$ are commonplace in condensed matter systems such fluids can thus not be described in terms of the standard Navier–Stokes equation or its relativistic counterpart because these are both boost invariant. This is of relevance to both critical and quantum critical systems, and our new theory is suitable to study the hydrodynamic behavior of (quantum) critical systems as well. We will here not attempt to classify all systems of this nature but it includes such vastly different phenomena as dynamic scaling in biology, see e.g. [19] for a recent example, quantum critical transport in strange metals, see e.g. [20], and viscous electron fluids [21].

## 2 Perfect fluids revisited

The equation of state of a perfect fluid with one conserved $U(1)$ current is of the form $P = P(T, \mu, v^2)$ where $T$ is the temperature, $\mu$ the chemical potential for the conserved charge density $n$ and where $v^2 = v^i v^i$ with $i = 1, \ldots, d$ is the velocity squared [1]. By charge we mean either electric charge, mass, baryon number or any other conserved particle species. The velocity is treated as a chemical potential conjugate to the momentum density $\mathscr{P}_i$. Since we assume rotational symmetries the momentum density must be proportional to $v^i$, i.e. $\mathscr{P}_i = \rho v^i$, where $\rho$ will be called the 'kinetic mass density'. When there is boost invariance one can define a rest frame temperature and chemical potential such that the equation of state for $P$ no longer depends on $v^2$. Otherwise there is an absolute frame with respect to which we measure $v^2$. From the equation of state one can determine the entropy density $s$, the charge density $n$ and the kinetic mass density $\rho$ via the Gibbs–Duhem relation $\delta P = s\delta T + n\delta\mu + \frac{1}{2}\rho\delta v^2$. Notice that $\mathscr{P}_i \delta v^i = \frac{1}{2}\rho\delta v^2$. The total energy density $\mathscr{E}$ is then given by the Euler relation $\mathscr{E} = Ts - P + v^i\mathscr{P}_i + \mu n$. We will frequently work with the internal energy $\tilde{\mathscr{E}} = \mathscr{E} - \rho v^2$. The first law is given by $\delta\tilde{\mathscr{E}} = T\delta s + \mu\delta n - \frac{1}{2}\rho\delta v^2$.

The perfect fluid energy-momentum tensor and $U(1)$ current in the laboratory (LAB) frame in which the fluid has velocity $v^i$ is given by [1]

$$T^0{}_0 = -\mathscr{E}, \quad T^0{}_j = \rho v^j, \quad T^i{}_0 = -(\mathscr{E} + P)v^i, \tag{1}$$

$$T^i{}_j = P\delta^i_j + \rho v^i v^j, \quad J^0 = n, \quad J^i = nv^i. \tag{2}$$

We note that this is not the most general form for these currents compatible with the symmetries. In general one can have five scalar quantities appearing in the energy-momentum tensor and two distinct scalars in the $U(1)$ current. However, in thermodynamic equilibrium all charges move with the same average velocity, which gives rise to the expression above. The energy momentum tensor (1) has the property that, using a coordinate transformation that takes the form of a Galilean boost, one can go to a moving coordinate system in which all the fluxes are zero and in which the charges are: $\mathscr{E} - v^i\mathscr{P}_i$ (the internal energy $\tilde{\mathscr{E}}$) for the energy density, $\mathscr{P}_i$ for momentum, $n$ for charge density and furthermore $T^i{}_j = P\delta^j_i$ in terms of the pressure.

---

[3]More precisely the no-go theorem tells us that fluids cannot have Galilean symmetries with $z \neq 2$.

[4]Hydrodynamics for $z = 2$ Schrödinger systems was considered from a modern perspective in e.g. Refs. [10–18].

Using (1), (2), the conservation equations $\partial_\mu T^\mu{}_\nu = 0$ and $\partial_\mu J^\mu = 0$ take the form

$$0 = \left(\partial_t + v^i \partial_i\right)\mathscr{E} + (\mathscr{E} + P)\partial_i v^i + v^i \partial_i P \,, \tag{3}$$

$$0 = \rho\left(\partial_t + v^i \partial_i\right)v^j + \partial_j P + v^j\left((\partial_t + v^i \partial_i)\rho + \rho\partial_i v^i\right), \tag{4}$$

$$0 = \left(\partial_t + v^i \partial_i\right)n + n\partial_i v^i \,, \tag{5}$$

where the second equation is the generalized Euler equation. Using these perfect fluid equations of motion together with the first law for $\delta\mathscr{E}$ it can be shown that entropy is conserved, i.e. $\left(\partial_t + v^i \partial_i\right)s + s\partial_i v^i = 0$.

In general there are two a priori different notions of mass density, namely the 'kinetic mass density' which is the proportionality between the momentum density and the velocity that we call $\rho$ and the 'substance mass density' which is the number of particles per unit volume that we denote by $n$ (assuming that we are dealing with a system that has a conserved particle number). We will see that for fluids with Galilean symmetry these two notions are equal to each other, $\rho = n$[5]. It is well known that in a relativistic theory $\rho$ is no longer conserved because the faster particles move the more massive they become. Whenever $\rho \neq n$, the Euler equation contains an extra force term $-v^j\left((\partial_t + v^i \partial_i)\rho + \rho\partial_i v^i\right)$ which is due to the non-conservation of $\rho$. We see here that in a general non-boost invariant setting we need to sharply distinguish between these two different notions of mass.

We will briefly investigate the role of scale symmetries. The most general scale symmetry, compatible with rotations, is of the form $t \to \lambda^z t$, $x^i \to \lambda x^i$ for time and space coordinates $t$ and $x^i$, where $z$ is the dynamical exponent. This implies the Ward identity $-zT^0{}_0 + T^i{}_i = 0$ which for a perfect fluid leads to the thermodynamic condition: $dP = z\tilde{\mathscr{E}} + (z-1)\rho v^2$.

In appendix A we briefly review how standard relativistic and non-relativistic fluids are recovered from the general description. In [1] it was shown that only $z = 1$ is compatible with Lorentz boost symmetry and that only $z = 2$ is compatible with Galilean boost symmetry, while for $z \neq 1, 2$, as shown in [1], we cannot have any boost symmetry which thus requires one to use the general formalism developed here and in [1,4].

To set the stage for the main analysis we start by studying the hydrodynamic modes associated with perturbations around a global thermal equilibrium where the fluctuations obey the perfect fluid equations. For simplicity we consider throughout this paper a global equilibrium at rest, i.e. with $v_0^i = 0$ where the 0 subscript refers to the unperturbed equilibrium configuration. We thus consider $\tilde{\mathscr{E}} = \tilde{\mathscr{E}}_0 + \delta\tilde{\mathscr{E}}$, $P = P_0 + \delta P$ and $v^i = \delta v^i$ with $\tilde{\mathscr{E}}_0$ and $P_0$ constants. The fluctuation equations that follow from (3)–(5) are

$$0 = \partial_t \delta\tilde{\mathscr{E}} + \left(\tilde{\mathscr{E}}_0 + P_0\right)\partial_i \delta v^i \,, \tag{6}$$

$$0 = \rho_0 \partial_t \delta v^i + \partial_i \delta P \,, \tag{7}$$

$$0 = \partial_t \delta n + n_0 \partial_i \delta v^i \,. \tag{8}$$

There are as many hydrodynamic modes as there are conservation equations. We will now first discuss these modes at the perfect fluid level. By going to Fourier space it follows that there are two propagating sound modes $\delta P$ and $\delta v_\parallel = \frac{k^i}{k}\delta v^i$ where $k^i$ is the momentum of the mode and $d$ non-propagating modes with dispersion relation $\omega = 0$. These are the $d - 1$ shear modes $\delta v_\perp^i = \left(\delta_j^i - \frac{k^i k^j}{k^2}\right)\delta v^j$ and the diffusion mode $\delta\frac{s}{n}$. This latter fact is easily seen by eliminating $\partial_i \delta v^i$ from (6) and (8) and using the first law $\delta\tilde{\mathscr{E}} = T_0 n_0 \delta\frac{s}{n} + \frac{\tilde{\mathscr{E}}_0 + P_0}{n_0}\delta n$ with $v_0^i = 0$.

---

[5]In the Galilean case, on dimensional grounds, we should have $\rho = mn$ where $m$ is the mass of the identical particles. We will absorb the constant $m$ into $n$.

The dispersion relation for the sound mode $\delta v_\parallel$ is $\omega = \pm v_s k$ and likewise for $\delta P$, where

$$v_s^2 = \frac{\tilde{\mathscr{E}}_0 + P_0}{\rho_0} \left(\frac{\partial P_0}{\partial \tilde{\mathscr{E}}_0}\right)_{n_0} + \frac{n_0}{\rho_0} \left(\frac{\partial P_0}{\partial n_0}\right)_{\tilde{\mathscr{E}}_0} = \frac{n_0}{\rho_0} \left(\frac{\partial P_0}{\partial n_0}\right)_{\frac{s_0}{n_0}} \tag{9}$$

is the speed of sound. The second equality follows from (49) by writing $\delta \frac{s}{n}$ in terms of variations of $P$ and $n$ and isolating $\delta P$. When the system is scale invariant with a generic dynamical exponent $z$, which from now on we will refer to as a Lifshitz fluid, and when $v_0^i = 0$, we have $dP_0 = z\tilde{\mathscr{E}}_0$, so that the speed of sound is given by

$$v_s^2 = \frac{z}{d} \frac{\tilde{\mathscr{E}}_0 + P_0}{\rho_0}. \tag{10}$$

The speed of sound for perturbations around a moving global equilibrium, i.e. with $v_0^i \neq 0$ is discussed in [1]. In the following we will compute the corrections to all hydrodynamic modes that originate when going away from perfect fluid behaviour. For definiteness we will assume the presence of a $U(1)$ current and associated thermodynamic variable $n$, but the case without $U(1)$ is easily obtained by ignoring the $n$-dependence in our results.

## 3 Hydrodynamic frame choice

Perfect fluids describe systems in local thermodynamic equilibrium. We will now assume that there is no local equilibrium anymore. We will do so in the usual sense of performing a derivative expansion around the perfect fluid. We will work to first order in derivatives and restrict to small fluctuations in the fluid variables, deferring a more general study to [4]. When we move away from local equilibrium we need to specify what we mean by the local fluid variables such as temperature and velocity. This is commonly known as a hydrodynamic frame choice. Once such a choice has been made one writes down the most general constitutive relations for the conserved currents as well as the entropy current. The free functions in the constitutive relations are restricted by demanding local positivity of entropy production, which requires a detailed study of the entropy current. We will perform this procedure below and consequently derive the allowed transport coefficients. We will subsequently study the effect of the new transport coefficients on the dispersion relations of the hydrodynamic modes.

The energy-momentum tensor has in general one and exactly one negative eigenvalue. We can use the associated eigenvector to provide us with a definition of the velocity that is valid at any order in the derivative expansion. The associated hydrodynamic frame is the well known Landau frame and it is defined by $T^\mu{}_\nu u^\nu = -\tilde{\mathscr{E}} u^\mu$, where the eigenvector is parameterized as $u^\mu = u^0(1, v^i)$ with $v^i$ the velocity of the fluid and where the eigenvalue $\tilde{\mathscr{E}}$ is the internal energy. The prefactor $u^0$ is not fixed by this condition. The Landau frame conditions provide $d + 1$ definitions that can be used to fix the hydrodynamic frame. We have in case of a U(1) symmetry, however, $d + 2$ fluid variables so we need one more frame condition. This can be taken to be $\bar{u}_\mu J^\mu = -\tilde{n} = -(u^0)^{-1} n$. However this requires a covariant version of $u^\mu$ which we have denoted as $\bar{u}_\mu$. Generally, we do not have a non-degenerate metric at our disposal to raise and lower indices. In general there seems to be no canonical choice for $\bar{u}_\mu$. For our purposes it will be convenient to choose $\bar{u}_\mu$ such that $\bar{u}_\mu$ obeys $u^\mu \bar{u}_\mu = -1$ and that it is parameterized entirely in terms of $v^i$. This means that it has the general form $\bar{u}_\mu = (u^0)^{-1}(-1 - Bv^2, Bv^i)$. Here, both $u^0$ and $B$ can depend on $v^2$. Again, $u^0$ is not fixed by this condition. For a relativistic fluid we take $B = (1 - v^2)^{-1}$ and $u^0 = (1 - v^2)^{-1/2}$ which is dictated by Lorentz covariance while for a Galilean fluid we take $B = 0$ and $u^0 = 1$ as follows from Galilean covariance. At first order in perturbations the choices for $u^0$ and $B$ are irrelevant. For convenience we set $u^0 = 1$ and $B = 0$.

# 4 Constitutive relations

In order to compute the hydrodynamic modes we consider a double expansion. We fluctuate around a global equilibrium at rest, i.e. with $v_0^i = 0$ where the 0 subscript refers to the unperturbed equilibrium configuration and we restrict to terms that are up to first order in derivatives.

In the Landau frame the linearized energy-momentum tensor and charge current (assuming the presence of a $U(1)$ current) up to first order in derivatives are given by

$$T^0{}_0 = -\tilde{\mathscr{E}}_0 - \delta\tilde{\mathscr{E}}, \qquad T^i{}_0 = -\left(\tilde{\mathscr{E}}_0 + P_0\right)\delta v^i, \tag{11}$$

$$T^0{}_j = \rho_0 \delta v^j + T^0_{(1)j}, \qquad T^i{}_j = (P_0 + \delta P)\delta^i_j + T^i_{(1)j}, \tag{12}$$

$$J^0 = n_0 + \delta n, \qquad J^i = n_0 \delta v^i + J^i_{(1)}, \tag{13}$$

$$T^0_{(1)j} = -\pi_0 \partial_t \delta v^j + T_0(\alpha_0 + \gamma_0)\partial_j \delta\frac{\mu}{T}, \tag{14}$$

$$T^i_{(1)j} = -\zeta_0 \delta_{ij}\partial_k \delta v^k - \eta_0\left(\partial_i \delta v^j + \partial_j \delta v^i - \frac{2}{d}\delta_{ij}\partial_k \delta v^k\right), \tag{15}$$

$$J^i_{(1)} = (\alpha_0 - \gamma_0)\partial_t \delta v^i - T_0\sigma_0 \partial_i \delta\frac{\mu}{T}, \tag{16}$$

where $\zeta_0$, $\eta_0$, $\gamma_0$, $\pi_0$, $\alpha_0$ and $\sigma_0$ are first order transport coefficients. We will see in the next section that there is one more transport coefficient, that we will denote by $a$, that appears in the analysis of the entropy current.

The expansion is an on-shell expansion meaning that the first order correction terms are the most general set of derivatives that are unrelated by the leading order equations of motion (3)–(5), which take the form

$$\partial_t \delta T = -T_0\left(\frac{\partial P_0}{\partial \tilde{\mathscr{E}}_0}\right)_{n_0}\partial_i \delta v^i, \tag{17}$$

$$\partial_i \delta T = -\frac{\rho_0 T_0}{\tilde{\mathscr{E}}_0 + P_0}\partial_t \delta v^i - \frac{T_0^2 n_0}{\tilde{\mathscr{E}}_0 + P_0}\partial_i \delta\frac{\mu}{T}, \tag{18}$$

$$\partial_t \delta\frac{\mu}{T} = -\frac{1}{T_0}\left(\frac{\partial P_0}{\partial n_0}\right)_{\tilde{\mathscr{E}}_0}\partial_i \delta v^i. \tag{19}$$

This way of writing the perfect fluid equations follows from (57) and the real space equivalents of equations (61) and (62) (obtained by replacing $\omega$ by $i\partial_t$ and $k^i$ by $-i\partial_i$). Hence, using the linearized leading order equations of motion we can express the derivatives of $\delta T$ and the time derivative of $\delta\frac{\mu}{T}$ in terms of derivatives of $\delta v^i$ and the spatial derivative of $\delta\frac{\mu}{T}$. The reason we choose this set of on-shell independent derivatives is explained in appendix B. Furthermore, the derivative expansion must be such that we allow for all terms that are consistent with $SO(d)$ rotational symmetry which means that all derivatives transform as scalars, vectors or tensors under $SO(d)$ and that the Ward identity $T^i{}_j = T^j{}_i$ is obeyed.

# 5 Entropy current and Onsager relations

In this section we will define and construct the entropy current. The entropy current $S^\mu$ consists of a canonical piece and a non-canonical piece, i.e. $S^\mu = S^\mu_{\text{can}} + S^\mu_{\text{non}}$. The canonical entropy current is defined as

$$S^\mu_{\text{can}} = -\frac{1}{\tilde{T}}T^\mu{}_\nu u^\nu + \frac{P}{\tilde{T}}u^\mu - \frac{\tilde{\mu}}{\tilde{T}}J^\mu, \tag{20}$$

with $\bar{u}_\mu S^\mu_{\text{can}} = -\tilde{s} = -(u^0)^{-1}s$, $\tilde{T} = u^0 T$ and $\tilde{\mu} = u^0\mu$. Since at the linearized order $u^0 = 1$ the distinction between $\tilde{T}$, $\tilde{\mu}$ and $T$, $\mu$ disappears so that we drop the tildes from now on.

For a perfect fluid the canonical entropy current is $su^\mu$. The non-canonical entropy current is the most general current constructed from the fluid variables that is at least first order in derivatives, so that it vanishes for a perfect fluid, and such that $\partial_\mu S^\mu \geq 0$ for all fluid configurations. Using the Gibbs–Duhem relation it follows that on-shell and in any hydrodynamic frame

$$\partial_\mu S^\mu_{\text{can}} = -\left(T^\mu{}_\nu - T^\mu_{(0)\nu}\right)\partial_\mu \frac{u^\nu}{T} - \left(J^\mu - J^\mu_{(0)}\right)\partial_\mu \frac{\mu}{T}, \tag{21}$$

where the (0) subscript denotes perfect fluid.

We will split the currents $T^\mu{}_\nu - T^\mu_{(0)\nu}$ and $J^\mu - J^\mu_{(0)}$ into a dissipative and a non-dissipative part $T^\mu{}_\nu - T^\mu_{(0)\nu} = T^\mu_{\text{D}\nu} + T^\mu_{\text{ND}\nu}$ and $J^\mu - J^\mu_{(0)} = J^\mu_{\text{D}} + J^\mu_{\text{ND}}$, to be defined below, where we require that the dissipative (D) and non-dissipative (ND) tensors are both symmetric in their spatial indices. Without this assumption new unphysical parameters that have no origin in the constitutive relations appear in $\partial_\mu S^\mu$. We define this decomposition by requiring

$$\partial_\mu S^\mu = -T^\mu_{\text{D}\nu}\partial_\mu \frac{u^\nu}{T} - J^\mu_{\text{D}}\partial_\mu \frac{\mu}{T} \geq 0. \tag{22}$$

This implies that the non-canonical entropy current satisfies

$$\partial_\mu S^\mu_{\text{non}} = T^\mu_{\text{ND}\nu}\partial_\mu \frac{u^\nu}{T} + J^\mu_{ND}\partial_\mu \frac{\mu}{T}. \tag{23}$$

For linearized perturbations the divergence of the entropy current is

$$\partial_\mu S^\mu = -\frac{1}{T}T^i_{(1)j}\partial_i \delta v^j - \frac{1}{T}T^0_{(1)j}\partial_t \delta v^j - J^i_{(1)}\partial_i \delta \frac{\mu}{T} + \partial_\mu S^\mu_{(1)\text{non}}, \tag{24}$$

where we used the linearized version of (21) as well as the constitutive relations (11)–(16) and where $S^\mu_{(1)\text{non}}$ is the first order correction to $S^\mu_{\text{non}}$.

In appendix B we show that the divergence of the non-canonical entropy current is

$$\begin{aligned}
\partial_\mu S^\mu_{(1)\text{non}} = & -\left[T_0 a_T \left(\frac{\partial P_0}{\partial \tilde{\mathcal{E}}_0}\right)_{n_0} + \frac{a_{\frac{\mu}{T}}}{T_0}\left(\frac{\partial P_0}{\partial n_0}\right)_{\tilde{\mathcal{E}}_0}\right](\partial_i \delta v^i)^2 \\
& + \frac{\rho_0 T_0}{\tilde{\mathcal{E}}_0 + P_0}a_T(\partial_t \delta v^i)^2 + \left[\frac{T_0^2 n_0}{\tilde{\mathcal{E}}_0 + P_0}a_T - a_{\frac{\mu}{T}}\right]\partial_i \delta \frac{\mu}{T}\partial_t \delta v^i,
\end{aligned} \tag{25}$$

where $a_T = \left(\frac{\partial a}{\partial T_0}\right)_{\frac{\mu_0}{T_0}}$ and $a_{\frac{\mu}{T}} = \left(\frac{\partial a}{\partial \frac{\mu_0}{T_0}}\right)_{T_0}$ are derivatives of a single transport coefficient $a$ that appears in $S^\mu_{(1)\text{non}}$.

Combining (25) with (24) in which we substitute the constitutive relations (15)–(16) we can now impose positive entropy production. The expression $\partial_\mu S^\mu \geq 0$ is a quadratic form on the space of derivatives $\partial_t \delta v^j$, $\partial_i \delta v^j$ and $\partial_i \delta \frac{\mu}{T}$. Demanding that the quadratic form is positive semi-definite for all fluid configurations leads to

$$\bar{\zeta}_0 \geq 0, \quad \eta_0 \geq 0, \quad \bar{\pi}_0 \geq 0, \quad \sigma_0 \geq 0, \quad \bar{\alpha}_0^2 \leq \bar{\pi}_0 \sigma_0, \tag{26}$$

where we defined

$$\zeta_0 = \bar{\zeta}_0 + a_T T_0^2 \left(\frac{\partial P_0}{\partial \tilde{\mathcal{E}}_0}\right)_{n_0} + a_{\frac{\mu}{T}}\left(\frac{\partial P_0}{\partial n_0}\right)_{\tilde{\mathcal{E}}_0}, \tag{27}$$

$$\pi_0 = \bar{\pi}_0 - a_T \frac{\rho_0 T_0^2}{\tilde{\mathcal{E}}_0 + P_0}, \tag{28}$$

$$\alpha_0 = \bar{\alpha}_0 + \frac{a_T}{2}\frac{n_0 T_0^2}{\tilde{\mathcal{E}}_0 + P_0} - \frac{a_{\frac{\mu}{T}}}{2}. \tag{29}$$

We have split the coefficients into a dissipative part (the non-negative barred coefficients) and a non-dissipative part depending on $a$. Notice in particular that there is no constraint on $\gamma_0$, appearing in (14) and (16), which hence does not produce entropy.

We can recover the known Lorentzian and Galilean boost invariant cases. In the Lorentzian case the momentum current is equal to the energy flux, $T^0{}_i = -T^i{}_0$. Equating these currents at first order in derivatives for the dissipative and non-dissipative parts separately tells us that $a = 0$ and that $\bar{\pi}_0 = 0$ as well as $\bar{\alpha}_0 = -\gamma_0$. From the last inequality in (26) we then find $\bar{\alpha}_0 = \gamma_0 = 0$. Likewise for a Galilean invariant fluid we have that the momentum density equals the particle flux, $T^0{}_i = J^i$. Following a similar argument we end up with $a = 0$, $\bar{\pi}_0 = -\alpha_0 + \gamma_0$ and $\sigma_0 = -\bar{\alpha}_0 - \gamma_0$, so that the last inequality of (26) sets $\gamma_0 = 0$. This latter result agrees with the linearized Landau frame expressions given in [15]. We conclude that the two coefficients $\gamma_0$ and $a$ vanish in the Lorentz and Galilean boost invariant cases.

The nature of two non-dissipative coefficients $\gamma_0$ and $a$ is rather different as we now discuss. Also it may appear puzzling that we end up with 6 transport coefficients even though we started with 5 in the constitutive relations. However in all known cases the coefficients in the non-canonical entropy current can be obtained from a hydrostatic partition function [22, 23] and therefore also appear in new terms in the constitutive relations once we turn on non-trivial background fields. A preliminary analysis suggest that the same will happen here but we defer further details to [4]. The nature of $\gamma_0$ can be made more precise by invoking the Onsager theorem which holds when the underlying theory of which we are making a fluid approximation is time reversal symmetric [24]. The Onsager theorem states that in this case the currents and the derivatives $\partial_t \delta v^j$, $\partial_i \delta v^j$ and $\partial_i \delta \frac{\mu}{T}$ appearing in $\partial_\mu S^\mu$ at the linearized level are related via a constant symmetric matrix. Different tensor structures do not mix and in our case the only tensors that do not automatically obey the Onsager theorem are the vectors. Hence we can focus on the 2 currents $T^0_{D(1)i}$ and $J^i_{D(1)}$ appearing on the right hand side of (22). Their constitutive relations can be expressed as

$$
\begin{pmatrix} T^0_{D(1)i} \\ J^i_{D(1)} \end{pmatrix} = \begin{pmatrix} -\bar{\pi}_0 & \bar{\alpha}_0 + \gamma_0 \\ \bar{\alpha}_0 - \gamma_0 & -\sigma_0 \end{pmatrix} \begin{pmatrix} \partial_t \delta v^i \\ T_0 \partial_i \delta \frac{\mu}{T} \end{pmatrix}.
\tag{30}
$$

By the Onsager theorem this 2 by 2 matrix must be symmetric. This tells us that $\gamma_0 = 0$ for systems with time reversal symmetry. The coefficient $\gamma_0$ is an example of what is called Berry transport in [25] which describes non-dissipative out of equilibrium transport[6].

We thus find 5 dissipative transport coefficients, which, as we will see, can be split into two viscosities $\bar{\zeta}_0$ and $\eta_0$ and three conductivities $\bar{\pi}_0$, $\bar{\alpha}_0$ and $\sigma_0$. On top of that there is 1 non-dissipative transport coefficient $a$. Next we will study their effect on the hydrodynamic modes.

# 6 Hydrodynamic modes

In order to study the dispersion relation for the hydrodynamic modes we analyzed the equations of motion $\partial_\mu T^\mu{}_\nu = 0$ and $\partial_\mu J^\mu = 0$ coming from (11)–(16) in appendix D (using results from appendix C). Equation (57) generalizes the linearized perturbations of the Navier–Stokes equations to the case without any boost symmetry. The main result relevant here is expressed in equations (67)–(69) together with (60) which describe the fluctuations of $\delta v^i_\perp$, $\delta v_\parallel$, $\delta P$ and $\delta \frac{s}{n}$.

Since we work up to first order in derivatives we can solve for the eigenfrequencies of these equations using a dispersion relation of the form $\omega = c_1 k - i c_2 k^2 + \mathcal{O}(k^3)$ with $c_1$ and $c_2$ real.

---

[6]We thank Kristan Jensen for pointing this out to us.

The eigenfrequencies of the equations for $\delta v_\perp^i$, $\delta v_\parallel$, $\delta P$ and $\delta \frac{s}{n}$ are commonly referred to as the shear, sound and diffusion modes and their dispersion relations and multiplicities up to first order for general non-boost invariant hydrodynamicsx are given by

$$\omega_{\text{shear}} = -i\frac{\eta_0}{\rho_0}k^2, \qquad \text{with multiplicity } d-1, \tag{31}$$

$$\omega_{\text{sound}} = \pm v_s k - i\Gamma k^2, \qquad \text{with multiplicity } 2, \tag{32}$$

$$\omega_{\text{diff}} = -i\frac{(\tilde{\mathcal{E}}_0 + P_0)^2}{n_0^3 T_0 c_P}\sigma_0 k^2, \qquad \text{with multiplicity } 1, \tag{33}$$

where $c_P$ is the specific heat at constant pressure defined in the second equation of (51). One sound mode and the diffusion mode are each one particular linear combination of $\delta v_\parallel$ and $\delta \frac{s}{n}$. Furthermore $\Gamma$ is the sound attenuation constant given by

$$\Gamma = \frac{1}{2\rho_0 v_s^2}\left[\left[\bar{\zeta}_0 + \frac{2}{d}(d-1)\eta_0\right]v_s^2 + \bar{\pi}_0 v_s^4 + \sigma_0\left(\left(\frac{\partial P_0}{\partial n_0}\right)_{\tilde{\mathcal{E}}_0}\right)^2 + 2\bar{\alpha}_0 v_s^2\left(\frac{\partial P_0}{\partial n_0}\right)_{\tilde{\mathcal{E}}_0}\right]. \tag{34}$$

This expression follows from (70) where we used that

$$\zeta_0 + \pi_0 v_s^2 + 2\alpha_0\left(\frac{\partial P_0}{\partial \tilde{\mathcal{E}}_0}\right)_{n_0} = \bar{\zeta}_0 + \bar{\pi}_0 v_s^2 + 2\bar{\alpha}_0\left(\frac{\partial P_0}{\partial \tilde{\mathcal{E}}_0}\right)_{n_0}. \tag{35}$$

This equation tells us, as expected, that the non-dissipative coefficient $a$ appearing in $S^\mu_{(1)\text{non}}$ does not contribute to the attenuation of the sound mode.

From (68) we see that $\frac{(\tilde{\mathcal{E}}_0 + P_0)^2}{n_0^3 T_0 c_P}\sigma_0$ is a diffusion constant for the diffusion of $\delta\frac{s}{n}$, the entropy per particle (or charge). The constant $\sigma_0$ is the associated charge or particle conductivity depending on whether the $U(1)$ current $J^\mu$ describes charge or particle conservation. From the condition of positive entropy production (26) we see that order $k^2$ terms in $\omega_{\text{shear}}$ and $\omega_{\text{diff}}$ have a negative imaginary part so these are damping terms. In appendix E we show that this is also true for the sound attenuation constant $\Gamma$.

Equation (74) allows us to define the thermal conductivity $\kappa_0$ via

$$\partial_\mu S^\mu = \kappa_0 \frac{1}{T_0^2}(\partial_i \delta T)^2 + \dots, \tag{36}$$

where the dots contain terms with spatial derivatives of $\delta\frac{\mu}{T}$ and $\delta v^i$. It then follows from (74) that the thermal conductivity is proportional to the transport coefficient $\bar{\pi}_0$ via

$$\kappa_0 = \frac{(\tilde{\mathcal{E}}_0 + P_0)^2}{\rho_0^2 T_0}\bar{\pi}_0. \tag{37}$$

We have thus established that $\bar{\pi}_0$ and $\sigma_0$ are thermal and charge/particle conductivity, respectively. Since $\bar{\alpha}_0$ appears in the same conductivity matrix (73) and since, due to the inequality $\bar{\alpha}_0^2 \leq \bar{\pi}_0\sigma_0$, it is nonzero only when $\bar{\pi}_0$ and $\sigma_0$ are both nonzero we will refer to it as the thermo-charge or thermo-particle conductivity.

In both the Lorentzian and Galilean cases the expressions for $\omega_{\text{shear}}$, $\omega_{\text{sound}}$ and $\omega_{\text{diff}}$ agree with textbook results [26, 27]. In the case of a Lorentz invariant system we have, due to the equality of momentum density and energy flux, that $\bar{\pi}_0 = \bar{\alpha}_0 = a = 0$ and $\rho_0 = \tilde{\mathcal{E}}_0 + P_0$, so that

$$\Gamma_{\text{Lor}} = \frac{1}{2(\tilde{\mathcal{E}}_0 + P_0)}\left[\zeta_0 + \frac{1}{d}(d-1)\eta_0\right] + \frac{\sigma_0}{2(\tilde{\mathcal{E}}_0 + P_0)v_s^2}\left(\left(\frac{\partial P_0}{\partial n_0}\right)_{\tilde{\mathcal{E}}_0}\right)^2. \tag{38}$$

In the case of a Galilean invariant system we have that the momentum density equals the mass flux from which we conclude that $\bar{\pi}_0 = -\bar{\alpha}_0 = \sigma_0$, $a = 0$ and of course $\rho_0 = n_0$. In this case, with the help of (55), we can write $\Gamma$ (for any arbitrary equation of state $P_0 = P_0(\tilde{\mathcal{E}}_0, n_0)$) as the well-known result

$$\Gamma_{\text{Gal}} = \frac{1}{2}\frac{\zeta_0}{n_0} + \frac{1}{d}(d-1)\frac{\eta_0}{n_0} + \frac{1}{2}\frac{\kappa_0}{n_0}\left(\frac{1}{c_V} - \frac{1}{c_P}\right). \tag{39}$$

An important application of our results concerns Lifshitz hydrodynamics in which we have a Lifshitz scale symmetry with a dynamical exponent $z$. We have seen that fluids with $z \neq 1, 2$ cannot be boost invariant. When there is a scale symmetry we have (for $v_0^i = 0$) that $P_0 = P_0(\tilde{\mathcal{E}}_0)$ so that $\left(\frac{\partial P_0}{\partial n_0}\right)_{\tilde{\mathcal{E}}_0} = 0$. Furthermore, scale invariance implies that the bulk viscosity $\bar{\zeta}_0$ vanishes as well as the non-dissipative coefficient $a$ (which follows from demanding that $T^i_{\text{D}j}$ and $T^i_{\text{ND}j}$ are both traceless), so that the expression for $\Gamma$ for a Lifshitz invariant system with or without a $U(1)$ current simplifies to the novel expression

$$\Gamma_{\text{Lif}} = \frac{1}{d}(d-1)\frac{\eta_0}{\rho_0} + \frac{1}{2}\frac{\bar{\pi}_0}{\rho_0}v_s^2 = \frac{1}{d}(d-1)\frac{\eta_0}{\rho_0} + \frac{z}{2d}\frac{\tilde{\mathcal{E}}_0 + P_0}{\rho_0^2}\bar{\pi}_0, \tag{40}$$

where in the second equality we used (10). By measuring the speed of sound $v_s$ and the attenuation of the shear mode (31) we can determine $\frac{\bar{\pi}_0}{\rho_0}$ by measuring the attenuation $\Gamma$ of the sound mode.

# 7 Discussion

In a Galilean fluid the thermal conductivity $\kappa_0$ is proportional to $\sigma_0$ and is thus related to the diffusion of $\delta\frac{s}{n}$, the entropy per unit mass or particle. In a relativistic fluid the thermal conductivity $\kappa_0$ vanishes. What we have shown is that in general, when there are no boosts present, the thermal conductivity can be any positive number independent of the diffusion constant (which is proportional to $\sigma_0$). In fact even when there is no $U(1)$ current, so that $n = \mu = 0$, the fluid can still have a non-vanishing $\kappa_0$.

It is well known that sound attenuation in a conformal fluid is entirely controlled by the shear viscosity. An important difference between conformal and Lifshitz fluids is that in the latter case we see an additional contribution to the sound attenuation due to the non-vanishing of $\kappa_0$.

We have seen that for a generic non-boost invariant fluid there are 5 dissipative and 1 non-dissipative transport coefficients[7]. The 5 dissipative coefficients are $\bar{\zeta}_0$ (bulk viscosity), $\eta_0$ (shear viscosity), $\bar{\pi}_0 \propto \kappa_0$ (thermal conductivity), $\sigma_0$ (charge or particle conductivity which is proportional to the charge or particle diffusion constant), $\bar{\alpha}_0$ (thermo-charge or thermo-particle conductivity which is nonzero only when $\bar{\pi}_0$ and $\sigma_0$ are nonzero since $\bar{\alpha}_0^2 \leq \bar{\pi}_0\sigma_0$) and the non-dissipative coefficient $a$. It would be interesting to see if we can understand the presence of $a$ by constructing the hydrostatic partition function [22, 23] up to first order in derivatives.[8] After imposing scale invariance the 5+1 coefficients get reduced to 4 dissipative and 0 non-dissipative transport coefficients because scale invariance requires $\bar{\zeta}_0 = a = 0$. In the absence of a $U(1)$ current this is further reduced to 2 dissipative transport coefficients ($\eta_0$ and $\bar{\pi}_0$).

In [4] we will study the general properties of all first order transport coefficients and not just those that survive after restricting ourselves to small fluctuations around a fluid at rest.

---

[7]When the theory of which we are making a fluid approximation is not time reversal invariant there is a second non-dissipative transport coefficient that we denoted by $\gamma_0$.

[8]The leading order hydrostatic partition function for general perfect fluids is given in [1].

This will give rise to a generalization of the Navier–Stokes equation for systems without boost symmetries. For applications to systems with spontaneous symmetry breaking without boost symmetries it would be very worthwhile to generalize our discussion to include superfluids. It would furthermore be interesting to consider non-boost invariant analogues of the fluid approach to magnetohydrodynamics developed in [28,29].

Finally, it would be interesting to study the properties of the first order transport terms in holographic realizations of systems with Lifshitz thermodynamics such as in [30–34] and to see if they obey any universal properties such as the $\eta/s$ bound derived for holographic systems dual to strongly coupled conformal fluids in [35] (in this connection see also [16,17,36,37]). More generally, it would be important to develop a full-fledged fluid/gravity correspondence for Lifshitz systems, in analogy with the conformal relativistic [38,39] and non-relativistic ($z = 2$ Schrödinger) case [10–13].

**Acknowledgements.**

We thank Alexander Abanov, Jay Armas, Blaise Goutéraux, Elias Kiritsis, Koenraad Schalm, Henk Stoof and Lárus Thorlacius for useful discussions. We especially thank Sašo Grozdanov and Kristan Jensen for careful reading of this manuscript and for the many useful discussions. All the authors thank Nordita for hospitality and support during the 2016 workshop "Black Holes and Emergent Spacetime". The work of NO is supported in part by the project "Towards a deeper understanding of black holes with non-relativistic holography" of the Independent Research Fund Denmark (grant number DFF-6108-00340). JH and NO gratefully acknowledge support from the Simons Center for Geometry and Physics, Stony Brook University at which some of the research for this paper was performed during the 2017 workshop "Applied Newton-Cartan Geometry". JH acknowledges hospitality of the Niels Bohr Institute and NO acknowledges hospitality of the Universities of Amsterdam and Utrecht during part of this work. This work was further supported by the Netherlands Organisation for Scientific Research (NWO) under the VICI grant 680-47-603, and the Delta-Institute for Theoretical Physics (D-ITP) that is funded by the Dutch Ministry of Education, Culture and Science (OCW).

## A Boost invariant perfect fluids

We briefly discuss how standard Lorentz and Galilean invariant situations are recovered from the general perfect fluid defined in section 2. For relativistic fluids we have the Lorentz boost Ward identity $T^i{}_0 = -T^0{}_i$, so that $\rho = \mathscr{E} + P = \gamma^2(\tilde{\mathscr{E}} + P)$, where $\gamma = (1 - v^2)^{-1/2}$ is the Lorentz contraction factor. We redefine $T = \gamma^{-1}\tilde{T}$, $s = \gamma\tilde{s}$, $\mu = \gamma^{-1}\tilde{\mu}$ and $n = \gamma\tilde{n}$, so that we recover the standard thermodynamic relations $\tilde{\mathscr{E}} + P = \tilde{T}\tilde{s} + \tilde{\mu}\tilde{n}$ and $\delta\tilde{\mathscr{E}} = \tilde{T}\delta\tilde{s} + \tilde{\mu}\delta\tilde{n}$. In other words we have succeeded in removing the dependence of the equation of state $P = P(\tilde{T}, \tilde{\mu})$ on $v^2$, as expected in a boost invariant setting. This independence of $P$ on $v^2$ is only compatible with scale invariance if we set $z = 1$ in agreement with the known fact that we can only add scale symmetries to the Poincaré algebra for $z = 1$.

For non-relativistic Galilean fluids the boost Ward identity reads $T^0{}_i = J^i$ which implies $\rho = n$. If we define $\hat{\mathscr{E}} = \tilde{\mathscr{E}} + \frac{1}{2}nv^2$ and $\hat{\mu} = \mu + \frac{1}{2}v^2$, it follows that $\hat{\mathscr{E}} = Ts - P + \hat{\mu}n$ and $\delta\hat{\mathscr{E}} = T\delta s + \hat{\mu}\delta n$, where $\hat{\mathscr{E}}$ is the internal energy. In this case the scale Ward identity can be written as $dP = z\hat{\mathscr{E}} + \frac{z-2}{2}nv^2$. Since $P$ is a function of only $T$ and $\hat{\mu}$ and since $s = \left(\frac{\partial P}{\partial T}\right)_{\hat{\mu}}$ and $n = \left(\frac{\partial P}{\partial \hat{\mu}}\right)_T$ and because $\hat{\mathscr{E}} = \hat{\mathscr{E}}(s, n)$ it follows that the combination $(dP - z\hat{\mathscr{E}})/n$ is a function of only $T$ and $\hat{\mu}$ and not of $v^2$. We conclude that this is compatible with scale symmetries only for $z = 2$. On an algebraic level, in the case of the Bargmann algebra, we can add scale symmetries with general $z$ leading to the Schrödinger algebra with general $z$. We thus see

that we cannot form a perfect fluid with $z \neq 2$. Conversely, this means that whenever we are dealing with a physical system that is scale invariant with a dynamical exponent $z \neq 1, 2$, and that allows for a fluid description, we need to use the formalism of non-boost invariant fluid dynamics developed here and in [1, 4].

## B Non-canonical entropy current

In this appendix we will consider the constitutive relation for the non-canonical entropy current. We see that the first three terms on the right hand side of (24) are second order in perturbations. The ansatz for the non-canonical entropy current must therefore be written up to first order in derivatives and up to second order in fluctuations and is parameterized by

$$
S^0_{(1)\text{non}} = a_1 \partial_i \delta v^i + a_2 \delta v^i \partial_t \delta v^i + a_3 \delta v^i \partial_i \delta \frac{\mu}{T}, \tag{41}
$$

$$
S^i_{(1)\text{non}} = b_1 \partial_t \delta v^i + b_2 \delta v^j \partial_i \delta v^j + b_3 \delta v^j \partial_j \delta v^i + b_4 \delta v^i \partial_j \delta v^j + b_5 \partial_i \delta \frac{\mu}{T}, \tag{42}
$$

where the coefficients depend at most linearly on the fluctuations $\delta T$ and $\delta \frac{\mu}{T}$.

When writing down the constitutive relations for the conserved current we chose to work with the on-shell independent set of derivatives $\partial_t \delta v^j$, $\partial_i \delta v^j$ and $\partial_i \delta \frac{\mu}{T}$. The reason for this particular choice of derivatives is because these are the derivatives that appear in the divergence of the canonical part of the entropy current (24). In this way when expanding the conserved currents in derivatives of $\partial_t \delta v^j$, $\partial_i \delta v^j$ and $\partial_i \delta \frac{\mu}{T}$ we do not have to explicitly use the leading order equations of motion to convert from one set of derivatives to another. This is particularly advantageous when going beyond leading order perturbations [4]. This advantage does not apply to the non-canonical part of the entropy current $\partial_\mu S^\mu_{(1)\text{non}}$ (24) because it involves the derivatives $\partial_t \delta T$, $\partial_i \delta T$ and $\partial_t \delta \frac{\mu}{T}$ that need to be converted into the set $\partial_t \delta v^j$, $\partial_i \delta v^j$ and $\partial_i \delta \frac{\mu}{T}$ with the help of the leading order equations of motion.

The divergence $\partial_\mu S^\mu_{(1)\text{non}}$ contains terms that are of the form $\mathcal{O}(\partial^2)$ that can never be non-negative and terms that are of the form $\mathcal{O}(\partial)\mathcal{O}(\partial)$. Since the first three terms in (24) only contain terms of the latter form we must demand that $\partial_\mu S^\mu_{(1)\text{non}}$ only contains products of first order derivatives and no genuine second order derivatives. This leads to $a_1 = -b_1$, $a_2 = a_3 = b_2 = b_5 = 0$ and $b_3 = -b_4$, so that

$$
\partial_\mu S^\mu_{(1)\text{non}} = \partial_t a_1 \partial_i \delta v^i - \partial_i a_1 \partial_t \delta v^i + b_3 \left( \partial_i \delta v^j \partial_j \delta v^i - \partial_i \delta v^i \partial_j \delta v^j \right). \tag{43}
$$

The coefficient $b_3$ must be zero because otherwise we cannot fulfil (23) with $T^\mu_{\text{ND}\nu}$ symmetric in its spatial indices. We will rename $a_1 = a$ and to first order in fluctuations it is expanded as $a = a_0 + a_T \delta T + a_{\frac{\mu}{T}} \delta \frac{\mu}{T}$ where $a_T = \left( \frac{\partial a}{\partial T_0} \right)_{\frac{\mu_0}{T_0}}$ and $a_{\frac{\mu}{T}} = \left( \frac{\partial a}{\partial \frac{\mu_0}{T_0}} \right)_{T_0}$ and where the constant $a_0$ is unphysical because it does not appear in $\partial_\mu S^\mu_{(1)\text{non}}$. Using (17) to eliminate derivatives of $\delta T$ and time derivatives of $\delta \frac{\mu}{T}$ we obtain

$$
\partial_\mu S^\mu_{(1)\text{non}} = -\left[ T_0 a_T \left( \frac{\partial P_0}{\partial \tilde{\mathscr{E}}_0} \right)_{n_0} + \frac{a_{\frac{\mu}{T}}}{T_0} \left( \frac{\partial P_0}{\partial n_0} \right)_{\tilde{\mathscr{E}}_0} \right] (\partial_i \delta v^i)^2
$$
$$
+ \frac{\rho_0 T_0}{\tilde{\mathscr{E}}_0 + P_0} a_T (\partial_t \delta v^i)^2 + \left[ \frac{T_0^2 n_0}{\tilde{\mathscr{E}}_0 + P_0} a_T - a_{\frac{\mu}{T}} \right] \partial_i \delta \frac{\mu}{T} \partial_t \delta v^i. \tag{44}
$$

This is used in the main text to derive (25).

# C  Thermodynamic relations

We collect here a set of useful thermodynamic relations among derivatives of various thermodynamic quantities. For notational convenience we drop here the subscript 0. Throughout we set $v^i = 0$.

Consider the transformation from $(T, \frac{\mu}{T})$ to $(\tilde{\mathcal{E}}, n)$: $\begin{pmatrix} \delta\tilde{\mathcal{E}} \\ \delta n \end{pmatrix} = J \begin{pmatrix} \delta T \\ \delta\frac{\mu}{T} \end{pmatrix}$, where $J$ is the Jacobian. From inverting the Jacobian we learn that

$$\begin{pmatrix} \left(\frac{\partial T}{\partial \tilde{\mathcal{E}}}\right)_n & \left(\frac{\partial T}{\partial n}\right)_{\tilde{\mathcal{E}}} \\ \left(\frac{\partial \frac{\mu}{T}}{\partial \tilde{\mathcal{E}}}\right)_n & \left(\frac{\partial \frac{\mu}{T}}{\partial n}\right)_{\tilde{\mathcal{E}}} \end{pmatrix} = (\det J)^{-1} \begin{pmatrix} \left(\frac{\partial n}{\partial \frac{\mu}{T}}\right)_T & -\left(\frac{\partial \tilde{\mathcal{E}}}{\partial \frac{\mu}{T}}\right)_T \\ -\left(\frac{\partial n}{\partial T}\right)_{\frac{\mu}{T}} & \left(\frac{\partial \tilde{\mathcal{E}}}{\partial T}\right)_{\frac{\mu}{T}} \end{pmatrix}, \tag{45}$$

where

$$\det J = \left(\frac{\partial \tilde{\mathcal{E}}}{\partial T}\right)_{\frac{\mu}{T}} \left(\frac{\partial n}{\partial \frac{\mu}{T}}\right)_T - \left(\frac{\partial \tilde{\mathcal{E}}}{\partial \frac{\mu}{T}}\right)_T \left(\frac{\partial n}{\partial T}\right)_{\frac{\mu}{T}}. \tag{46}$$

By writing $P$ as a function of $(\tilde{\mathcal{E}}, n)$ and varying $(\tilde{\mathcal{E}}, n)$ and by writing $P$ as a function of $(T, \frac{\mu}{T})$ again varying $(\tilde{\mathcal{E}}, n)$, using the inverse $J^{-1}$ Jacobian, we find

$$\left(\frac{\partial \frac{\mu}{T}}{\partial \tilde{\mathcal{E}}}\right)_n = \frac{1}{nT}\left(\frac{\partial P}{\partial \tilde{\mathcal{E}}}\right)_n - \frac{\tilde{\mathcal{E}} + P}{nT^2}\left(\frac{\partial T}{\partial \tilde{\mathcal{E}}}\right)_n, \tag{47}$$

$$\left(\frac{\partial \frac{\mu}{T}}{\partial n}\right)_{\tilde{\mathcal{E}}} = \frac{1}{nT}\left(\frac{\partial P}{\partial n}\right)_{\tilde{\mathcal{E}}} - \frac{\tilde{\mathcal{E}} + P}{nT^2}\left(\frac{\partial T}{\partial n}\right)_{\tilde{\mathcal{E}}}. \tag{48}$$

By rewriting the first law we obtain for the entropy per unit charge $s/n$,

$$\delta\left(\frac{s}{n}\right) = \frac{1}{Tn}\delta\tilde{\mathcal{E}} - \frac{\tilde{\mathcal{E}} + P}{Tn^2}\delta n. \tag{49}$$

Using the equality of mixed second order derivatives of $s/n$ under interchanging the order and after using (47) we obtain the Maxwell relation

$$\left(\frac{\partial \frac{\mu}{T}}{\partial \tilde{\mathcal{E}}}\right)_n = \frac{1}{T^2}\left(\frac{\partial T}{\partial n}\right)_{\tilde{\mathcal{E}}} = \frac{1}{nT}\left(\frac{\partial P}{\partial \tilde{\mathcal{E}}}\right)_n - \frac{\tilde{\mathcal{E}} + P}{nT^2}\left(\frac{\partial T}{\partial \tilde{\mathcal{E}}}\right)_n. \tag{50}$$

For fluids that have a conserved particle number it is possible to define a specific heat. In the following we denote by $n$ the number of particles per unit volume. We define the specific heats $c_V$ and $c_P$ as

$$c_V = \frac{1}{n}\left(\frac{\partial \tilde{\mathcal{E}}}{\partial T}\right)_n = T\left(\frac{\partial \frac{s}{n}}{\partial T}\right)_n, \quad c_P = T\left(\frac{\partial \frac{s}{n}}{\partial T}\right)_P. \tag{51}$$

In order to derive a few useful relations involving $c_V$ and $c_P$ consider the change of variables from $(\tilde{\mathcal{E}}, n)$ to $(P, T)$: $\begin{pmatrix} \delta P \\ \delta T \end{pmatrix} = J'\begin{pmatrix} \delta\tilde{\mathcal{E}} \\ \delta n \end{pmatrix}$, where $J'$ is the Jacobian. We can use the inverse Jacobian $J'^{-1}$ to rewrite (49) as a variation with respect to $\delta P$ and $\delta T$. Using the expression for the components of $J'^{-1}$ in terms of the components of $J'$ we then find the following expression for the specific heat at constant pressure $c_P$,

$$c_P = -\frac{\rho v_s^2}{n^2}\left(\det J'\right)^{-1}, \tag{52}$$

where we used (9) and where we have

$$\det J' = \left(\frac{\partial P}{\partial \tilde{\mathscr{E}}}\right)_n \left(\frac{\partial T}{\partial n}\right)_{\tilde{\mathscr{E}}} - \left(\frac{\partial P}{\partial n}\right)_{\tilde{\mathscr{E}}} \left(\frac{\partial T}{\partial \tilde{\mathscr{E}}}\right)_n. \tag{53}$$

By computing the determinant of the inverse Jacobian $J^{-1}$ and using (47) and (48) we obtain

$$\det J = -nT(\det J')^{-1} = \frac{n^3 T}{\rho v_s^2} c_P, \tag{54}$$

where in the second equality we used (52).

Using (53) and (52) we can derive an expression for $\left(\frac{\partial T}{\partial n}\right)_{\tilde{\mathscr{E}}}$. The second equality in (50) gives another expression for $\left(\frac{\partial T}{\partial n}\right)_{\tilde{\mathscr{E}}}$. Eliminating $\left(\frac{\partial T}{\partial n}\right)_{\tilde{\mathscr{E}}}$ from these two expressions leads to

$$\left[\left(\frac{\partial P}{\partial \tilde{\mathscr{E}}}\right)_n\right]^2 = \frac{\rho v_s^2}{Tn}\left(\frac{1}{c_V} - \frac{1}{c_P}\right). \tag{55}$$

From which we derive the usual inequality $c_P \geq c_V$.

## D   Linearized perturbations

We will consider linearized perturbations that obey the Onsager theorem, i.e. for which $\gamma_0 = 0$. The linearized equations of motion $\partial_\mu T^\mu{}_\nu = 0$ and $\partial_\mu J^\mu = 0$ that follow from (11)–(16) are

$$0 = \partial_t \delta\tilde{\mathscr{E}} + \left(\tilde{\mathscr{E}}_0 + P_0\right)\partial_i \delta v^i, \tag{56}$$

$$0 = \rho_0 \partial_t \delta v^j + \partial_j \delta P - \pi_0 \partial_t^2 \delta v^j + T_0 \alpha_0 \partial_t \partial_j \delta\frac{\mu}{T} - \left(\zeta_0 + \frac{d-2}{d}\eta_0\right)\partial_j \partial_i \delta v^i - \eta_0 \partial_i \partial_i \delta v^j, \tag{57}$$

$$0 = \partial_t \delta n + n_0 \partial_i \delta v^i + \alpha_0 \partial_t \partial_i \delta v^i - T_0 \sigma_0 \partial_i \partial_i \delta\frac{\mu}{T}, \tag{58}$$

where $\zeta_0$, $\eta_0$ are the bulk and shear viscosity respectively. The coefficients $\pi_0$ and $\sigma_0$ are the thermal and charge/particle conductivity respectively. The coefficients $\pi_0$ and $\alpha_0$ both multiply terms that describe variations of the fluid's acceleration $\partial_t \delta v^i$. We note that this term can be written in terms of gradients of $\delta T$ and $\delta\frac{\mu}{T}$ via the leading order equation of motion (18).

We perform a Fourier transformation by taking $\delta\tilde{\mathscr{E}} = e^{-i\omega t + i\vec{k}\cdot\vec{x}}\delta\tilde{\mathscr{E}}(\omega, k)$ etc. and we define a velocity $\delta v_\parallel = \frac{k^i}{k}\delta v^i$ parallel to $k^j$ and a velocity $\delta v_\perp^i = \left(\delta_j^i - \frac{k^i k^j}{k^2}\right)\delta v^j$ perpendicular to $k^j$. The Fourier transform of (57) projected along $k^j$ gives

$$0 = \omega\rho_0 \delta v_\parallel - k\delta P + i\left(\zeta_0 + \frac{2}{d}(d-1)\eta_0\right)k^2 \delta v_\parallel + i\omega^2 \pi_0 \delta v_\parallel + iT_0 \alpha_0 \omega k\delta\frac{\mu}{T} = 0, \tag{59}$$

whereas projected orthogonal to $k^j$ we find

$$\rho_0 \omega \delta v_\perp^i + i\eta_0 k^2 \delta v_\perp^i + i\pi_0 \omega^2 \delta v_\perp^i = 0. \tag{60}$$

For $v_0^i = 0$ we can consider $\delta\tilde{\mathscr{E}}$, $\delta P$ and $\delta n$ as functions of $\delta T$ and $\delta\frac{\mu}{T}$ and derive a set of equations for the fluid variables $\delta T$, $\delta\frac{\mu}{T}$ and $\delta v^i$. We multiply (56) with $\left(\frac{\partial n_0}{\partial\frac{\mu_0}{T_0}}\right)_{T_0}$ and multiply (58) with $\left(\frac{\partial\tilde{\mathscr{E}}_0}{\partial\frac{\mu_0}{T_0}}\right)_{T_0}$ and subtract the two equations to obtain (after using (45) and (46) as well as the second equality in (50)),

$$0 = \omega\delta T - T_0\left(\frac{\partial P_0}{\partial\tilde{\mathscr{E}}_0}\right)_{n_0} k\delta v_\parallel + i\alpha_0\left(\frac{\partial T_0}{\partial n_0}\right)_{\tilde{\mathscr{E}}_0}\omega k\delta v_\parallel + iT_0\sigma_0\left(\frac{\partial T_0}{\partial n_0}\right)_{\tilde{\mathscr{E}}_0} k^2\delta\frac{\mu}{T}. \tag{61}$$

Similarly we multiply (56) with $\left(\frac{\partial n_0}{\partial T_0}\right)_{\frac{\mu_0}{T_0}}$ and multiply (58) with $\left(\frac{\partial \tilde{\mathscr{E}}_0}{\partial T_0}\right)_{\frac{\mu_0}{T_0}}$ and subtract the two equations leading to (after using (45) and (46)),

$$
0 \;=\; \omega\delta\frac{\mu}{T} - \frac{1}{T_0}\left(\frac{\partial P_0}{\partial n_0}\right)_{\tilde{\mathscr{E}}_0} k\delta v_\parallel + i\alpha_0\left(\frac{\partial \frac{\mu_0}{T_0}}{\partial n_0}\right)_{\tilde{\mathscr{E}}_0}\omega k\delta v_\parallel + iT_0\sigma_0\left(\frac{\partial \frac{\mu_0}{T_0}}{\partial n_0}\right)_{\tilde{\mathscr{E}}_0} k^2\delta\frac{\mu}{T}. \quad (62)
$$

Using the Gibbs–Duhem relation $\delta P = \frac{\tilde{\mathscr{E}}_0+P_0}{T_0}\delta T + T_0 n_0 \delta\frac{\mu}{T}$ it follows that equations (59), (60), (61) and (62) form a set of equations for the fluid variables $\delta T$, $\delta\frac{\mu}{T}$ and $\delta v^i$.

In order to study the hydrodynamic modes it will be convenient to derive a set of equations for the alternative set of fluid variables given by $\delta P$, $\delta\frac{s}{n}$ and $\delta v^i$. We can express $\delta\frac{s}{n}$ in terms of $\delta T$ and $\delta\frac{\mu}{T}$ as follows

$$
\begin{aligned}
\delta\frac{s}{n} \;&=\; \frac{n_0 c_P}{T_0\rho_0 v_s^2}\left(\frac{\partial P_0}{\partial n_0}\right)_{\tilde{\mathscr{E}}_0}\delta T - \frac{T_0 n_0 c_P}{\rho_0 v_s^2}\left(\frac{\partial P_0}{\partial \tilde{\mathscr{E}}_0}\right)_{n_0}\delta\frac{\mu}{T}\\
&=\; \frac{n_0 c_P}{(\tilde{\mathscr{E}}_0+P_0)\rho_0 v_s^2}\left(\frac{\partial P_0}{\partial n_0}\right)_{\tilde{\mathscr{E}}_0}\delta P - \frac{T_0 n_0 c_P}{\tilde{\mathscr{E}}_0+P_0}\delta\frac{\mu}{T},
\end{aligned} \quad (63)
$$

where the first equality follows from expressing $\delta\frac{s}{n}$ in (49) in terms of variations with respect to $\delta T$ and $\delta\frac{\mu}{T}$ which can be rewritten using equations (45), (50) and (54) into the above expression. The second equality follows from the Gibbs–Duhem relation and the expression for the speed of sound. By using (63) we can eliminate $\delta\frac{\mu}{T}$ from equation (59) in favor of $\delta P$ and $\delta\frac{s}{n}$ leading to

$$
\begin{aligned}
0 \;=\; & \omega\rho_0\delta v_\parallel - k\delta P + i\left(\zeta_0 + \frac{2}{d}(d-1)\eta_0\right)k^2\delta v_\parallel + i\omega^2\pi_0\delta v_\parallel\\
& -i\alpha_0\frac{\tilde{\mathscr{E}}_0+P_0}{n_0 c_P}\omega k\delta\frac{s}{n} + i\alpha_0\frac{1}{\rho_0 v_s^2}\left(\frac{\partial P_0}{\partial n_0}\right)_{\tilde{\mathscr{E}}_0}\omega k\delta P.
\end{aligned} \quad (64)
$$

We combine an appropriate linear combination of (61) and (62) to find

$$
\begin{aligned}
0 \;=\; & \omega\delta P - \rho_0 v_s^2 k\delta v_\parallel + i\alpha_0\left(\frac{\partial P_0}{\partial n_0}\right)_{\tilde{\mathscr{E}}_0}\omega k\delta v_\parallel\\
& -i\sigma_0\frac{\tilde{\mathscr{E}}_0+P_0}{n_0 c_P}\left(\frac{\partial P_0}{\partial n_0}\right)_{\tilde{\mathscr{E}}_0}k^2\delta\frac{s}{n} + i\frac{\sigma_0}{\rho_0 v_s^2}\left[\left(\frac{\partial P_0}{\partial n_0}\right)_{\tilde{\mathscr{E}}_0}\right]^2 k^2\delta P,
\end{aligned} \quad (65)
$$

where we used (9) as well as (63). By taking another appropriate linear combination of (61) and (62) we obtain

$$
0 \;=\; \omega\delta\frac{s}{n} + i\sigma_0\frac{(\tilde{\mathscr{E}}_0+P_0)^2}{T_0 n_0^3 c_P}k^2\delta\frac{s}{n} - i\alpha_0\frac{\tilde{\mathscr{E}}_0+P_0}{T_0 n_0^2}\omega k\delta v_\parallel - i\sigma_0\frac{\tilde{\mathscr{E}}_0+P_0}{T_0 n_0^2\rho_0 v_s^2}\left(\frac{\partial P_0}{\partial n_0}\right)_{\tilde{\mathscr{E}}_0}k^2\delta P, (66)
$$

where we used (48) to remove $\left(\frac{\partial \frac{\mu_0}{T_0}}{\partial n_0}\right)_{\tilde{\mathscr{E}}_0}$ from (62), and where equations (50), (53), (54) were used to simplify the result and finally we used (63) to express $\delta\frac{\mu}{T}$ at order $k^2$ in terms of $\delta\frac{s}{n}$ and $\delta P$.

We have thus managed to write the Fourier transform of the perturbation equations (56)–(58) in terms of $\delta v_\perp^i$, $\delta v_\parallel$, $\delta P$ and $\delta\frac{s}{n}$ leading to (60), (63), (64) and (65). Equation (64) can be simplified by multiplying it with $\frac{\omega}{\rho}$ and using equations (65) and (66) keeping only

terms that are linear in transport coefficients (because that is the order to which our results are valid). In a similar manner equations (66) and (65) can be simplified. This leads to

$$0 = \left(\omega^2 - v_s^2 k^2 + i\frac{\pi_0}{\rho_0}\omega\left(\omega^2 - v_s^2 k^2\right) + 2i\Gamma\omega k^2\right)\delta v_\parallel - i\sigma_0\frac{\tilde{\mathcal{E}}_0 + P_0}{\rho_0 n_0 c_P}\left(\frac{\partial P_0}{\partial n_0}\right)_{\tilde{\mathcal{E}}_0} k^3 \delta\frac{s}{n}, \quad (67)$$

$$0 = \left(\omega + i\frac{(\tilde{\mathcal{E}}_0 + P_0)^2}{n_0^3 T_0 c_P}\sigma_0 k^2\right)\delta\frac{s}{n} - i\frac{\tilde{\mathcal{E}}_0 + P_0}{T_0 n_0^2 v_s^2}\left[\sigma_0\left(\frac{\partial P_0}{\partial n_0}\right)_{\tilde{\mathcal{E}}_0} + \alpha_0 v_s^2\right]\omega k\delta v_\parallel, \quad (68)$$

$$0 = \left(\omega^2 - v_s^2 k^2 + 2i\Gamma\omega k^2\right)\delta P, \quad (69)$$

where $\Gamma$ is given by

$$\Gamma = \frac{1}{2\rho_0 v_s^2}\left[\left[\zeta_0 + \frac{2}{d}(d-1)\eta_0\right]v_s^2 + \pi_0 v_s^4 + \sigma_0\left(\left(\frac{\partial P_0}{\partial n_0}\right)_{\tilde{\mathcal{E}}_0}\right)^2 + 2\alpha_0 v_s^2\left(\frac{\partial P_0}{\partial n_0}\right)_{\tilde{\mathcal{E}}_0}\right]. \quad (70)$$

We see that the equation for $\delta P$ is decoupled. Equations (67)–(69) together with (60) are the main result of this appendix and are used to obtain (31)–(34) in the main text.

## E  Positivity of the sound attenuation constant

In order to show that $\Gamma$ given in (34) is non-negative when (26) is obeyed we complete a square leading to

$$\Gamma = \frac{1}{2}\frac{\bar{\zeta}_0}{\rho_0} + \frac{1}{d}(d-1)\frac{\eta_0}{\rho_0} + \frac{1}{\rho_0}\left(\bar{\alpha}_0 \pm \sqrt{\bar{\pi}_0\sigma_0}\right)\left(\frac{\partial P_0}{\partial n_0}\right)_{\tilde{\mathcal{E}}_0}$$
$$+ \frac{1}{2\rho_0 v_s^2}\left(\sqrt{\bar{\pi}_0}v_s^2 \mp \sqrt{\sigma_0}\left(\frac{\partial P_0}{\partial n_0}\right)_{\tilde{\mathcal{E}}_0}\right)^2. \quad (71)$$

The upper sign is for when $\left(\frac{\partial P_0}{\partial n_0}\right)_{\tilde{\mathcal{E}}_0} \geq 0$ and the lower sign for when $\left(\frac{\partial P_0}{\partial n_0}\right)_{\tilde{\mathcal{E}}_0} \leq 0$. Since $-\sqrt{\bar{\pi}_0\sigma_0} \leq \bar{\alpha}_0 \leq \sqrt{\bar{\pi}_0\sigma_0}$ we see that all terms on the right hand side are non-negative.

Another way to see that the $\bar{\pi}_0$, $\bar{\alpha}_0$ and $\sigma_0$ terms make a non-negative contribution to $\Gamma$ for any arbitrary equation of state is to write (34) as

$$2\rho_0 v_s^2 \Gamma = \left[\bar{\zeta}_0 + \frac{2}{d}(d-1)\eta_0\right]v_s^2$$
$$+ \frac{(\tilde{\mathcal{E}}_0 + P_0)^2}{\rho_0^2}\left[\left(\frac{\partial P_0}{\partial\tilde{\mathcal{E}}_0}\right)_{n_0}\right]^2\bar{\pi}_0 + 2\left(\bar{\pi}_0 + \frac{\rho_0}{n_0}\bar{\alpha}_0\right)\frac{\tilde{\mathcal{E}}_0 + P_0}{\rho_0}\left(\frac{\partial P_0}{\partial\tilde{\mathcal{E}}_0}\right)_{n_0}\frac{n_0}{\rho_0}\left(\frac{\partial P_0}{\partial n_0}\right)_{\tilde{\mathcal{E}}_0}$$
$$+ \left(\bar{\pi}_0 + 2\frac{\rho_0}{n_0}\bar{\alpha}_0 + \frac{\rho_0^2}{n_0^2}\sigma_0\right)\frac{n_0^2}{\rho_0^2}\left[\left(\frac{\partial P_0}{\partial n_0}\right)_{\tilde{\mathcal{E}}_0}\right]^2 \quad (72)$$

and to view the last 3 terms as the quadratic form

$$\begin{pmatrix} \bar{\pi}_0 & \bar{\pi}_0 + \frac{\rho_0}{n_0}\bar{\alpha}_0 \\ \bar{\pi}_0 + \frac{\rho_0}{n_0}\bar{\alpha}_0 & \bar{\pi}_0 + 2\frac{\rho_0}{n_0}\bar{\alpha}_0 + \frac{\rho_0^2}{n_0^2}\sigma_0 \end{pmatrix} \quad (73)$$

on the space of derivatives of the equation of state, i.e. $\frac{\tilde{\mathcal{E}}_0 + P_0}{\rho_0}\left(\frac{\partial P_0}{\partial\tilde{\mathcal{E}}_0}\right)_{n_0}$ and $\frac{n_0}{\rho_0}\left(\frac{\partial P_0}{\partial n_0}\right)_{\tilde{\mathcal{E}}_0}$. The quadratic form is positive semi-definite due to (26).

The same matrix appears in the expression for the divergence of the entropy current when expressed as a quadratic form on the space of gradients of $\delta T$ and $\delta \frac{\mu}{T}$, i.e.

$$
\begin{aligned}
T_0 \partial_\mu S^\mu &= \begin{pmatrix} \frac{\tilde{\mathscr{E}}_0 + P_0}{\rho_0 T_0} \partial_i \delta T & \frac{n_0}{\rho_0} T_0 \partial_i \delta \frac{\mu}{T} \end{pmatrix} \begin{pmatrix} \bar{\pi}_0 & \bar{\pi}_0 + \frac{\rho_0}{n_0} \bar{\alpha}_0 \\ \bar{\pi}_0 + \frac{\rho_0}{n_0} \bar{\alpha}_0 & \bar{\pi}_0 + 2\frac{\rho_0}{n_0} \bar{\alpha}_0 + \frac{\rho_0^2}{n_0^2} \sigma_0 \end{pmatrix} \\
&\quad \times \begin{pmatrix} \frac{\tilde{\mathscr{E}}_0 + P_0}{\rho_0 T_0} \partial_i \delta T \\ \frac{n_0}{\rho_0} T_0 \partial_i \delta \frac{\mu}{T} \end{pmatrix} - T^i_{(1)j} \partial_i \delta v^j ,
\end{aligned} \tag{74}
$$

where we used (24), (25) as well as (18) to replace $\partial_t \delta v^i$ derivatives in terms of gradients of $\delta T$ and $\delta \frac{\mu}{T}$.

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
