# Peer review of "Hydrodynamic Modes of Homogeneous and Isotropic Fluids"

_SciPost Physics, doi:SciPost Phys. 5, 014 (2018)_

## Round 1 · Referee Report · Anonymous (Referee 1) · 2018-3-14

Strengths

  1. The paper is clearly written
  2. The results are new and generalise older known results

Weaknesses

  1. It is not clear how useful these results are and whether we can gain new physical insight from them.

Report

This paper discusses transport in general theories of hydrodynamics which are homogenous and isotropic but not necessarily Lorentz invariant. The authors find new transport coefficients for such theories, not present in Galilean or Lorentz invariant hydrodynamics. They carry out a linearised analysis and find how the new transport coefficients affect physical quantities such as the diffusion constant or dispersion relations of the fluid. The results are novel. What is perhaps missing is whether or not we can expect these to affect physical observables in a realistic or semi-realistic system.

Requested changes

  1. (At the authors discretion) Add some text on how these results affect realistic physical systems. Something more refined then, say, the second paragraph of the current introduction.

  • validity: top
  • significance: good
  • originality: good
  • clarity: top
  • formatting: excellent
  • grammar: perfect

Author:  Niels Obers  on 2018-05-13  [id 252]

(in reply to Report 1 on 2018-03-14)

We thank the referees for their useful comments. In line with this the following corrections have been made:

1) In the 4th paragraph of the introduction we have added some sentences stressing that already at the perfect fluid level (as shown in Ref.[1]) there are now results (such as novel expressions for the speed of sound), which may be observable.

2) In regard concrete applications we have added the very recent Ref. [21] (on viscous electron fluid ) which in fact is a concrete example of non-boost invariant system to which our framework applies. (This is mentioned at the very end of the Introduction, section 1) ref. [21] = R. J. Doornenbal, M. Polini, and R. A. Duine, \Spin-vorticity coupling in viscous electron Fluids,"ArXiv e-prints (Mar., 2018) , arXiv:1803.03549 [cond-mat.mes-hall].

3) We have added in the 2nd paragraph of Section 2 more details on the form of the perfect fluid energy momentum tensor.

4) We have also updated the citation Ref.[9] with publication data.

We hope to herewith have addressed the referees remarks satisfactorily.

---

## Round 1 · Referee Report · Anonymous (Referee 2) · 2018-3-30

Strengths

The systematic investigation of the consequences of lack of boost invariance in hydrodynamic systems is of great interest theoretically.

Weaknesses

The presentation is rather technical, maybe not so transparent for people not in the field

Report

This paper derives the constitutive relations for first-order transport in the generic fluid that does not have any symmetry other than translation and a global charge,
continuing the study set up in

http://arxiv.org/abs/1710.04708v2

That first paper contains much of the discussion related to the motivation for this study, while this paper is mostly devoted to the linearised fluctuation analysis,
and the study of the constraints on the transport coefficients that arise from requiring positive entropy production, following the treatment that has become standard by now.
Still, the results are interesting enough, and in the event that the theory will find applications to some real system, it is useful to have them at hand, and it may also help to elucidate the general properties of hydrodynamics.

One point of confusion that may arise concerning the validity of this theory is the following: as the authors explain in the introduction, this is to be applied to a fluid that could have microscopic boost invariance, but broken by a medium. However if the fluid interacts with this medium, it is only the total energy-momentum tensor that is conserved. So in general one would expect the boost-breaking effects to be of the same order as the energy-nonconservation effect. Under what assumptions or conditions can one separate the two?

Requested changes

1) While the section 2 contains a description of the perfect fluid, presumably in order to make the paper self-contained, I found that insufficient and I had to look at the companion paper to understand this part. I suggest that here the authors give more details on how the stress energy tensor in (2.1), (2.2) is derived, in particular why the form given is not the most general compatible with the assumed symmetries.

2) In the introduction the authors state that they find “new hydrodynamic modes”, what do they mean here? the modes listed in section 6 are the shear, sound, and diffusion, like in the conventional case.

---

## Round 2 · Referee Report · Anonymous (Referee 1) · 2018-5-30

Report

The authors have addressed my concerns. I approve of publication in this journal.

---

## Round 2 · Referee Report · Anonymous (Referee 2) · 2018-6-22

Report

I am satisfied with the changes, and approve the resubmitted version.

---

## Round 2 · List of Changes

1) In the 4th paragraph of the introduction we have added some sentences stressing that already at the perfect fluid level (as shown in Ref.[1]) there are now results (such as novel expressions for the speed of sound), which may be observable.

2) In regard concrete applications we have added the very recent Ref. [21] (on viscous electron fluid) which in fact is a concrete example of non-boost invariant system to which our framework applies. (This is mentioned at the very end of the Introduction, section 1)

3) We have added in the 2nd paragraph of Section 2 more details on the form of the perfect fluid energy momentum tensor.

4) We have also updated the citation Ref.[9] with publication data.

---

## Editorial Decision

published